# Development of An Innovative and Quick Method for the Isolation of *Clostridium botulinum* Strains Involved in Avian Botulism Outbreaks

**DOI:** 10.3390/toxins12010042

**Published:** 2020-01-10

**Authors:** Thibault Le Gratiet, Typhaine Poezevara, Sandra Rouxel, Emmanuelle Houard, Christelle Mazuet, Marianne Chemaly, Caroline Le Maréchal

**Affiliations:** 1Unit of Hygiene and Quality of Poultry and Pork Products, French Agency for Food, Environmental and Occupational Health & Safety (ANSES), BP 53, 22440 Ploufragan, France; thibault.legratiet@anses.fr (T.L.G.);; 2UFR of Life Sciences and Environment, University of Rennes 1, 35 000 Rennes, France; 3National Reference Center for Anaerobic Bacteria and Botulism, Institut Pasteur, 25-28 rue du Docteur Roux, 75724 Paris, France

**Keywords:** *Clostridium botulinum*, isolation, botulism

## Abstract

Avian botulism is a serious neuroparalytic disease mainly caused by a type C/D botulinum neurotoxin produced by *Clostridium botulinum* group III, one of the entwined bacterial species from the *Clostridium novyi sensu lato* genospecies. Its isolation is very challenging due to the absence of selective media and the instability of the phage carrying the gene encoding for the neurotoxin. The present study describes the development of an original method for isolating *C. botulinum* group III strains. Briefly, this method consists of streaking the InstaGene matrix extraction pellet on Egg Yolk Agar plates and then collecting the colonies with lipase and lecithinase activities. Using this approach, it was possible to isolate 21 *C. novyi sensu lato* strains from 22 enrichment broths of avian livers, including 14 toxic strains. This method was successfully used to re-isolate type C, D, C/D, and D/C strains from liver samples spiked with five spores per gram. This method is cheap, user-friendly, and reliable. It can be used to quickly isolate toxic strains involved in avian botulism with a 64% success rate and *C. novyi sensu lato* with a 95% rate. This opens up new perspectives for *C. botulinum* genomic research, which will shed light on the epidemiology of avian botulism.

## 1. Introduction

Botulism is a flaccid paralytic disease that can affect both humans and animals. Clinical signs are caused by the action of botulinum neurotoxins (BoNTs) on the neuromuscular junction at peripheral nerves [1,2], blocking the fusion of acetylcholine vesicles and thus preventing transmission of nerve impulses [3]. BoNTs are metalloproteases composed of a heavy chain that mediates binding to receptors and translocation across endosomal membranes and a light chain that cleaves one of the three “Soluble N-ethylmaleimide-Sensitive-Factor Attachment Protein Receptors” (SNARE), namely vesicle-associated membrane protein (VAMP), synaptosomal-associated protein 25 (SNAP-25), and syntaxin-1A/1B [2,4]. Proteolysis of SNARE by BoNTs prevents the fusion of synaptic vesicles with the neuronal presynaptic plasma membrane leading to the failure of neurotransmission [1].

BoNTs are currently represented by at least seven serotypes and more than 40 subtypes [5]. They are produced during vegetative growth by spore-forming and anaerobic Gram-positive bacteria of the genus Clostridium [1], in particular by *Clostridium botulinum*. BoNT-like genes have also been computationally identified in non-clostridial species such as *Weissella*, *Enterococcus,* and *Chryseobacterium* [6,7].

*C. botulinum* is divided into four distinct groups (I to IV). Most outbreaks of animal botulism are caused by BoNT C and D or a chimeric fusion of C and D, i.e., C/D or D/C, produced by *C. botulinum* group III strains [8]. Animal botulism is considered an emerging disease in Europe [9], especially in livestock where an outbreak can induce high mortality rates (up to 100% in a flock), resulting in large economic losses [9,10]. Moreover, considering spore persistence in the environment [11], outbreaks frequently recur in affected flocks [12]. Botulism can also have a huge impact on wildlife species, in which massive outbreaks have been reported worldwide [13] and yearly [10]. In Europe, avian botulism is frequently associated with BoNT C/D [14,15,16], while bovine botulism is associated with BoNT D/C or C [14,16]. Several factors may explain why BoNT responsible for botulism outbreaks varies among animal species. BoNT C/D is more lethal for chickens than BoNT C, which may explain why BoNT C/D and not BoNT C is related to avian botulism [17]. It has also been suggested that type C and C/D strains could occupy different ecological niches: mammals for BoNT C (mink, horses, cattle) and birds for BoNT C/D [16]. Chickens are considered naturally resistant to type D botulism but can serve as a reservoir [18], while bovines appear to be highly sensitive to BoNT D/C [19].

Despite being reported worldwide in the literature since the early twentieth century, animal botulism has only been partially studied and is still difficult to control. Efforts have been made to develop diagnostic tools [14,20,21,22,23,24] but many aspects of the disease still need to be addressed, notably the epidemiology of animal botulism, which is still poorly understood, and the development of molecular epidemiological tools that are sorely lacking. While epidemiological tools are available for *C. botulinum* group I [25,26], few have been developed for *C. botulinum* group III. One of the reasons behind this situation is the difficulty in isolating *C. botulinum* group III and the absence of any consensual and reliable isolation method for this pathogen. 

*C. botulinum* group III grow poorly in mixed culture and on agar plates. It is an obligatory anaerobic bacteria and there are no known traits suitable for selective cultivation [27]. Another problem in isolating toxigenic strains is the unstable lysogeny of the phage carrying the gene encoding for BoNT C, D, C/D, and D/C [28,29] that is easily lost under laboratory conditions [14], especially after several transfers in culture media [30]. The mechanisms behind the interactions between the phage carrying the *bont* gene and *C. botulinum* group III or *Clostridium novyi sensu lato* and factors inducing its loss are currently unknown. 

The conventional method used to isolate *C. botulinum* group III consists of repeatedly culturing the sample in pre-reduced Tryptone, Peptone, Glucose Yeast extract (TPGY) [27] or Fortified-Cooked Meat Medium (F-CMM) [30,31,32] and then streaking the enrichment broth on non-selective media such as Egg Yolk Agar (EYA) [32], Blood Agar Base No.2 (BAB2) with 5% defibrinated sheep blood and 2.5% agarose [32], or a McClung Toabe Agar plate [27]. Colonies positive for lipase (an iridescent sheen is visible on the surface of the colonies) and lecithinase (revealed as an opaque precipitate surrounding the colony) have to be collected and screened individually to detect the presence of BoNT genes or the colony’s ability to produce BoNTs. Lecithinase activity varies between strains, however, and some isolates have been reported as negative for lecithinase activity [33]. A positive lipase reaction alone does not distinguish *C. botulinum* GIII from other *C. botulinum* types or from other *Clostridium* species [30]. Moreover, it is extremely difficult to distinguish toxigenic from nontoxigenic strains, although it is highly recommended to collect the smallest colonies [32]. This method is time-consuming, requires specific skills and lots of practice, and a simpler and more efficient isolation method is much needed [27]. Anza et al. recently developed a method based on immunomagnetic separation using beads coated with antibodies targeting *C. botulinum* group III strains that allowed the isolation of both toxigenic and nontoxigenic strains. This approach, however, involves extra costs and ethical concerns for the production of antibodies. Alternative strategies are needed for the isolation of strains involved in avian botulism outbreaks.

*C. botulinum* group III is closely related to *Clostridium novyi* and *Clostridium haemolyticum* [26,34]. These bacterial species are historically defined by their ability to cause a certain disease i.e., botulism, black disease, and bacillary hemoglobinuria, respectively. Genomic and mass spectrometric comparison of these three bacterial species have shown that they are very similar or even indistinguishable [16,29,34]. The ‘‘genospecies’’ name of *C. novyi sensu lato* was proposed to include these entwined species [29]. The differences between them are related to their plasmid content and phages—notably, genes encoding for toxins—which upon horizontal movement can transfer disease-causing properties [34]. *Bont* gene translocation from a *C. botulinum* group III strain to a *C. novyi* strain, for instance, has already been demonstrated [31]. BoNT is responsible for the clinical signs observed in botulism. The usual objective when studying botulism is therefore, quite reasonably, to isolate BoNT-producing strains. However, considering the very close relatedness of toxic and nontoxic strains (i.e., strains with or without the *bont* gene), it would appear beneficial to isolate *C. novyi senu lato* strains instead of focusing only on *C. botulinum* group III when studying animal botulism. This strategy was applied by Anza et al. who considered both toxic and non-toxic *C. novyi senu lato* during the isolation of clostridial strains from samples collected from different botulism outbreaks [35].

This article presents a new user-friendly, quick, and reliable method for isolating both *C. botulinum* group III and *C. novyi sensu lato* strains involved in avian botulism outbreaks. This method opens new perspectives for the exploration of many topics, including the monitoring of strain dissemination or, more generally, strain tracking.

## 2. Results

### 2.1. Selection of Samples for Evaluating the Isolation Protocol

Between April 2013 and October 2019, 153 samples were confirmed as being positive for *Clostridium botulinum* group III by the French National Reference Laboratory (NRL) for avian botulism. 

As described in Figure 1a, 58.16% of avian botulism outbreaks concerned poultry production and 41.84% concerned wild avian species. *C. botulinum* group III mainly affects three avian families or species: 37.91% were wild Anatidae, 27.45% were turkeys, and 18.95% were chickens. The remaining 15.68% were other avian species either from poultry production (11.76%) or the wildlife sector (3.92%).

As shown in Figure 1b, 80.39% of these samples were of type C/D, 7.84% were of type D/C (only in turkeys), 3.27% were of type D (only in turkeys), and 8.50% indicated the presence of several *bont* genes at the same time. Indeed, some samples were simultaneously positive for more than one type of *C. botulinum* group III. No pure type C outbreak was detected over this period. It is noteworthy that for two outbreaks in wild birds, a signal was detected simultaneously for types C/D and E in 2018. Our study did not evaluate the isolation of *C. botulinum* type E.

Twenty-two positive enrichment broths (Figure 2a) were selected from the samples collected between 2013 and 2019 to represent the outbreaks diagnosed in France over this period, i.e., various BoNT types, various regions (as depicted in Figure 2b), as well as various avian species.

### 2.2. Evaluation of the Method Using Naturally Contaminated Samples

An original protocol was designed to isolate *C. botulinum* group III strains (Figure 3) and evaluated using the 22 naturally contaminated samples listed in Figure 2 and selected as previously explained.

Using this approach, it was possible to isolate toxic or non-toxic *C. novyi sensu lato* strains from 21 enrichment broths (examples of strains plated on EYA during the isolation process are shown in Figure 4) with a success rate of 95%. Fourteen toxic strains (64% of all the strains isolated) were successfully isolated from the tested enrichment broths. The isolates were still toxic after the isolation process by PCR and using the mouse bioassay. The remaining strains (32%) were positive for *C. novyi sensu lato*, showing that the method is appropriate for isolating strains in this clostridial group.

### 2.3. Confirmation of the Method Using Spiked Samples

Besides the assay with the method depicted in Figure 3 using naturally contaminated samples, samples spiked with a known initial spore contamination level and of a known BoNT type were used to confirm that the method is indeed appropriate for isolating *C. botulinum* group III strains involved in avian botulism outbreaks. A method had been previously developed and validated for detecting *C. botulinum* in avian liver using PCR with a view of diagnosing avian botulism [24]. Its limit of detection was determined as five spores per gram of liver. In our study, broiler livers were spiked with five spores per gram of *C. botulinum* type C, D, C/D, and D/C. The isolation process depicted in Figure 3 was successfully used to re-isolate the strains used to spike the livers (Table 1). This result proves that this protocol can be applied to isolate *C. botulinum* group III strains, whatever the BoNT type, with an initial concentration as low as five spores per gram of liver.

## 3. Discussion

It is considered very difficult and thus a real challenge to isolate toxigenic *C. botulinum* group III [14,27,30]. This study presents a new method for the isolation of toxic and non-toxic *C. novyi sensu lato* strains from avian livers tested positive for *bont* gene using real-time PCR after their enrichment in TPGY. 

Using the conventional isolation method, namely streaking a broth positive for BoNT-producing *C. botulinum* group III on non-selective agar plates, a high proportion of isolates are reported to be nontoxigenic [27,30]. Franciosa et al. reported the isolation of 11 strains from eight different samples by selecting at least ten colonies per plate, revealing a low rate of toxigenic strains on plates (less than 14%) [30]. In our study, the number of toxigenic colonies (i.e., positive for the *bont* gene using real-time PCR) out of all the colonies collected on plates and positive for *C. novyi sensu lato* varies from 25% to 100% depending on the strains.

When tested on naturally contaminated samples, the approach developed by Anza et al. using beads coated with antibodies managed to isolate 15 strains from 46 samples [35]. All these strains were positive for the *C. novyi sensu lato* gene and only eight were toxigenic [35]. The method we developed, described herein, managed to isolate 21 strains from 22 samples, 14 of these strains being toxigenic. Our cheaper, user-friendly method was thus more effective in isolating toxigenic strains. 

Our approach is based on the ability of *C. botulinum* GIII to survive the InstaGene extraction protocol, which seems to eliminate most of the other bacterial contaminants that usually disturb the isolation process. The InstaGene matrix has previously been shown to fail to extract intracellular DNA from *Bacillus cereus* spores [36]. Moreover, the same study demonstrated that the majority of *B. cereus* spores appeared little or unaffected by the InstaGene matrix extraction protocol. Our results seem to indicate that this is also the case for *C. botulinum* GIII, which appears to survive the InstaGene matrix extraction protocol very well.

Liver was the only matrix tested here because it is used for routine confirmation of avian botulism in our laboratory [15,24]. Using the conventional method, isolation was reported to be easier from liver than from caeca [27]. No comparisons between the method’s efficiency according to the matrices used were made during this study, but the ability of *C. botulinum* to survive the InstaGene extraction protocol might also make it possible to use other matrices successfully. 

The InstaGene matrix is made up of negatively charged microscopic beads that trap the metal ions required as catalysts or cofactors in enzymatic reactions. This prevents extracted DNA being degraded by enzymes, which are blocked by the unavailability of catalysts and cofactors. Cell disruption is performed by incubation at 95 °C for eight min. Few studies have been conducted to evaluate the thermal resistance of *C. botulinum* group III isolates, but the available data show that type C spores of terrestrial strains can resist 101 °C for 14 min and marine strains up to 93°C for 18 min, although variations in thermal resistance between isolates have been reported [37].

In order to take into account the potential diversity of *C. botulinum* group III strains and evaluate the method’s efficiency in isolating strains during outbreaks, a variety of enrichment samples from the collection of the French National Reference Laboratory (NRL) for avian botulism were selected so as to be as representative as possible of the avian botulism outbreaks diagnosed in France. The authors therefore compiled an overview of the avian botulism outbreaks diagnosed since 2013 and the enrichment broths were selected on this basis so as to cover, insofar as possible, the avian species, geographical areas and BoNT types involved. As in most European countries [14,16,35,38], BoNT type C/D (80.39%) was the most frequently detected in avian botulism outbreaks. Type D/C (7.84%), type D (3.27%), and samples positive for several or all primers (C, D, C/D, and D/C) (8.50%) were detected far more scarcely. Type D/C had been reported in studies conducted in Italy, where it had been found either in the intestinal contents of a stork [16] or from coot, duck, or heron samples [38]. All the type D/C-positive samples in our study were detected in turkeys and not in wild birds. Samples positive for all primers had also been reported in some other avian outbreaks [38]. It appears that type D is uncommon in avian botulism outbreaks and has not been reported in other available studies [16,21,22,38]. No type C BoNTs were detected either in the samples analyzed during our study or in previously published studies that distinguished type C/D from type C BoNTs [14,16]. 

Poultry and wild birds can both be infected by botulism. Broilers (18.95%) and turkeys (27.45%) were the main poultry species, while mallards and more generally Anatidae (37.91%) were the main wild birds. This result is in accordance with the avian species recorded as being affected by botulism in other countries [10,13,14,16,39] except for turkeys, for which case reports are rare in the literature [40].

## 4. Conclusions

The isolation method developed in this study managed to successfully isolate 95% of *C. novyi sensu lato* strains, including 64% of toxigenic strains when testing type C/D, D, D/C, and C samples. Moreover, it is quicker, cheaper, and more user-friendly than previous methods. 

It provides an efficient, reliable, and easy-to-use protocol for isolating *C. botulinum* group III strains and opens up new avenues to better explore strain diversity, track strains during an outbreak, and improve knowledge of this bacterium’s physiology.

## 5. Materials and Methods

### 5.1. Samples

An overview of the outbreaks diagnosed by the French NRL for avian botulism was compiled to be able to select representative samples of situations encountered in the field in order to develop and evaluate the method. All outbreaks confirmed by the French NRL (clinical signs and a positive real-time PCR detection in samples collected from animals [24]) between 2013 and 2019 were listed and are presented in Figure 1. 

Twenty-two naturally contaminated livers were used in this study and are listed in Table 1. These samples were collected from animals in a context of suspected avian botulism in various regions of France, as well as in Belgium, between July 2017 and June 2019. 

According to a previously optimized method [24], livers (≤25 g) were ten-fold diluted in TPGY and incubated for at least 24 h at 37 °C in an anaerobic chamber (A35, Don Whitley distributed by Biomérieux, Bruz, France). DNA was extracted using 1 mL of this enrichment broth and BoNT genes were detected using real-time PCR. All the samples used in this study were positive for types C/D, D/C, or D. Enrichment broths were stored at below −18°C until isolation.

### 5.2. Confirmation of the Method Using Spiked Samples

Spore solutions from *C. botulinum* type C (strain CIP-109 785, from the collection housed at the Institut Pasteur, Paris, France), type D (strain CIP-105 256, Institut Pasteur, Paris, France), type C/D (B17LNRB4, ANSES, Ploufragan, France), and type D/C (B18LNRB3, ANSES, Ploufragan, France) were prepared and titrated using the five-tube most probable number (MPN) method as previously described [41]. Samples of chicken liver (5 g) that tested negative for the presence of *C. botulinum* using real-time PCR were inoculated with 5 MPN spores of each strain per g of liver. Spiked livers were then processed as described above for naturally contaminated samples.

### 5.3. InstaGene DNA Extraction

Enrichment broths (1 mL) were gently thawed, ten-fold diluted in TPGY, and incubated for at least 72 h in an anaerobic chamber (A35, Don Whitley distributed by Biomérieux, Bruz, France) at 37 °C. One milliliter of this enrichment broth was then incubated at 80 °C for 15 min to inactivate BoNT. After centrifuging at 12,850× *g* for one minute, the pellet was resuspended in 200 µL of InstaGene matrix (Bio-Rad, Marne-La-Coquette, France) and incubated at 56 °C for 30 min then at 95 °C for 8 min. A final centrifugation at 12,850× *g* for 3 min was performed to separate microbeads (in the residual pellet) from the DNA extract (in the supernatant). 

### 5.4. Real-Time PCR

The real-time PCR, primers, and probes in this study were used according to Woudstra et al. [42]. Real-time PCR with a Bio-Rad CFX96 thermal cycler (Bio-Rad, Marne-La-Coquette, France) was used to detect the *C. botulinum* BoNT gene and a gene characteristic of *C. novyi sensu lato* [42]. Each assay was performed in a total volume of 20 μL that contained 5 μL DNA template, 10 μL PerfeCTaqPCR ToughMix (Quantabio, distributed by VWR, Fontenay-sous-Bois, France), and a final concentration of 600 nM for primers and 400 nM for probes. The thermal profile consisted of 5 min at 95 °C, followed by 45 cycles of denaturation at 95 °C for 15 s and annealing/elongation at 55 °C for 30 s. Each PCR run included positive and negative controls for each target. A sample was considered positive when the Ct (Cycle threshold) was below 38.

### 5.5. Isolation on Egg Yolk Agar

The isolation protocol is fully depicted in Figure 3. The pellet obtained after DNA extraction using InstaGene matrix (Bio-Rad, Marne-la-Coquette, France) was homogenized using a vortex and 100 µL per plate was streaked over a pre-reduced Egg Yolk Agar (EYA) medium comprising 4% Blood Agar Base No 2 (Oxoid, Dardilly, France), 1.5% Bacto Agar (VWR, Fontenay-sous-Bois, France), and extemporaneously-prepared 10% egg yolk emulsion. Five colonies per plate positive for lipase and lecithinase activities [27] were individually collected after 24 h, 48 h, and 72 h of incubation at 37 °C in an anaerobic chamber, and cultured each time in 1 mL of TPGY broth at 37 °C in the anaerobic chamber (A35 Don Whitley distributed by Biomérieux, Bruz, France) for at least 24 h. DNA was extracted from 800 µL of each culture using the InstaGene matrix (Bio-Rad, Marne-la-Coquette, France) and the *ntnh* gene and a gene characteristic of *C. novyi sensu lato* [42] was detected using real-time PCR as previously described [24]. The remaining 200 µL was stored in the anaerobic chamber until PCR results were obtained.

### 5.6. Strain Storage

PCR results were analyzed so as to select the enrichment broth with the lowest Ct (i.e., presenting the greatest homogeneity) for both PCRs. The remaining 200 µL of the colony enrichment broth with the lowest Ct for the BoNT gene (*ntnh*) and the gene characteristic of *C. novyi sensu lato* were incubated in 10 mL of TPGY broth at 37 °C in the anaerobic chamber for at least 72 h. Supernatant was discarded after centrifugation (except 1 mL was stored for the mouse bioassay for BoNT detection) and the pellet was washed three times with physiological water. Finally, the pellet was resuspended in physiological water and stored first at a temperature below −18 °C and then at a temperature below −80 °C. Strain purity was checked by streaking 100 µL of the final suspension on pre-reduced EYA plates. After incubation for 24 h at 37 °C under anaerobic conditions, the EYA plates were examined to evaluate contamination.

### 5.7. Detection of BoNT Production by Strains Following Isolation

The detection of BoNT in the 24-h enrichment culture of isolated strains was based on the mouse lethality assay. The tests were performed in accordance with European Directive 2010/63/EU on the protection of animals used for scientific purposes (laboratory animal use agreement no. 2013-0116). One mL of enrichment broth was collected, centrifuged, filtered, and five-fold diluted in 50 mM phosphate buffer (pH 6.5) containing 1% gelatin. A volume of 0.5 mL was injected intraperitoneally into Swiss mice weighing 20–22 g (Charles River Laboratories, l’Arbresle, France). The mice were observed for up to 4 days for the presence of typical clinical signs (pinching of the waist, labored breathing, and paresis) and euthanized immediately after observation of such signs.

## Figures and Tables

**Figure 1 toxins-12-00042-f001:**
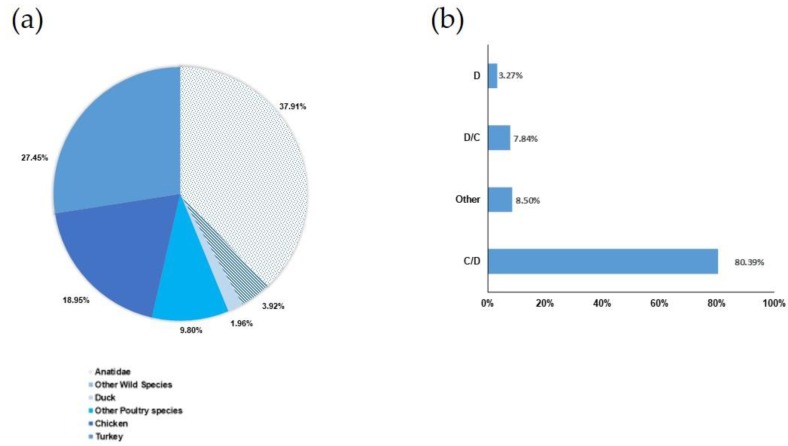
Overview of avian botulism outbreaks diagnosed by the French National Reference Laboratory (NRL) between 2013 and 2019. (**a**) Distribution of avian species (poultry species (58.2%) are represented by colors and wild birds (41.8%) by symbols); (**b**) BoNT types detected in the liver (C/D: detection of the *bont* C/D gene, D/C: detection of *bont* D/C, D: detection of *bont* D, other: simultaneous detection of several *bont* genes (out of C, C/D, D/C, and D)).

**Figure 2 toxins-12-00042-f002:**
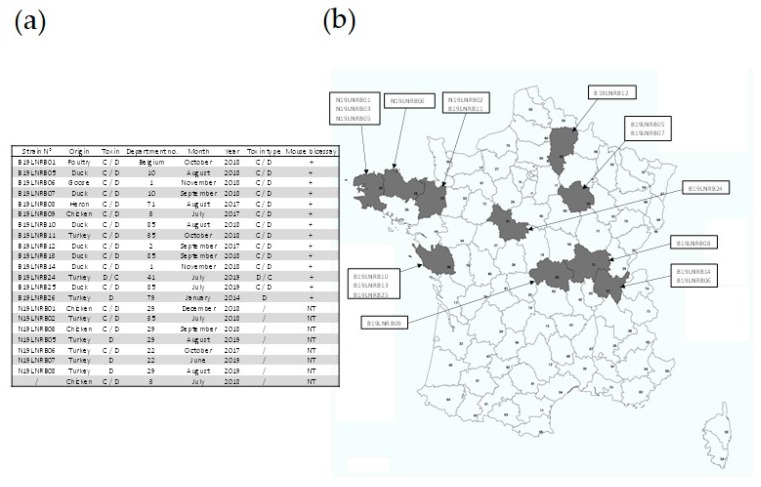
Data regarding the enrichment broths tested in our study to evaluate the isolation method. (**a**) Overview of the data (strain reference number, animal species, BoNT type identified using PCR, number of the *département* (French administrative unit) in which the outbreak occurred, month of the outbreak, year of the outbreak, BoNT type of the strain isolated, results of the mouse bioassay (+: positive, –: negative, NT: Not Tested), (**b**) Map showing the distribution of the outbreaks and enrichment broths included in this study.

**Figure 3 toxins-12-00042-f003:**
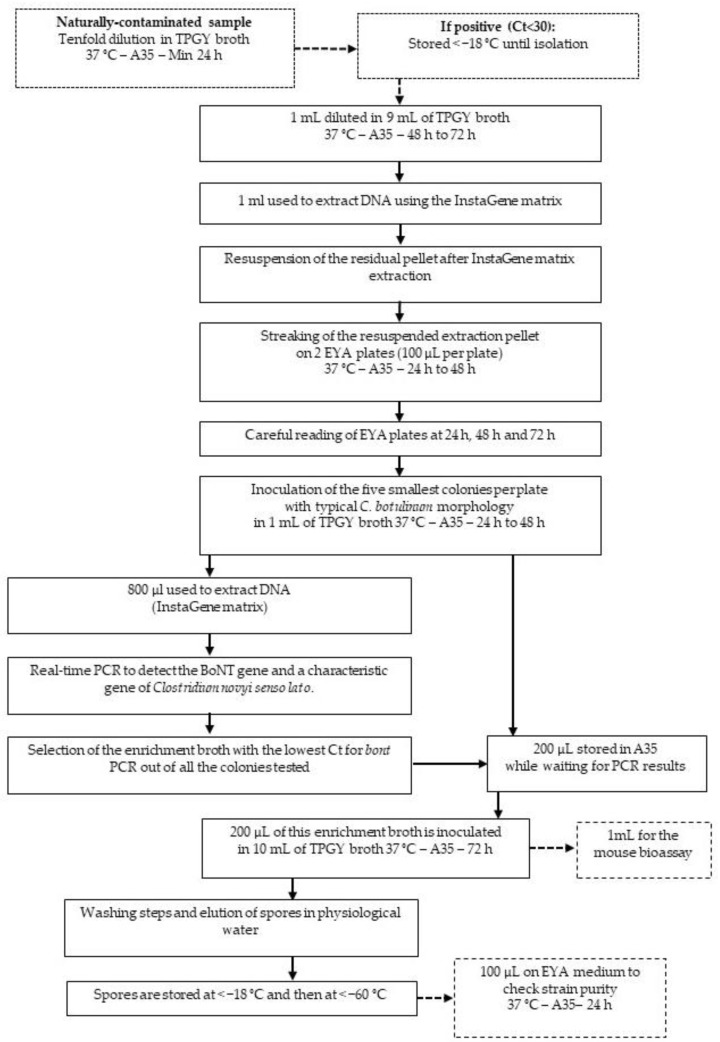
Workflow of *Clostridium novyi sensu lato* isolation from samples positive either for type C, C/D, D, D/C *bont* genes using real-time PCR. (A35: anaerobic station A35, Ct: Cycle threshold detected using real-time PCR).

**Figure 4 toxins-12-00042-f004:**
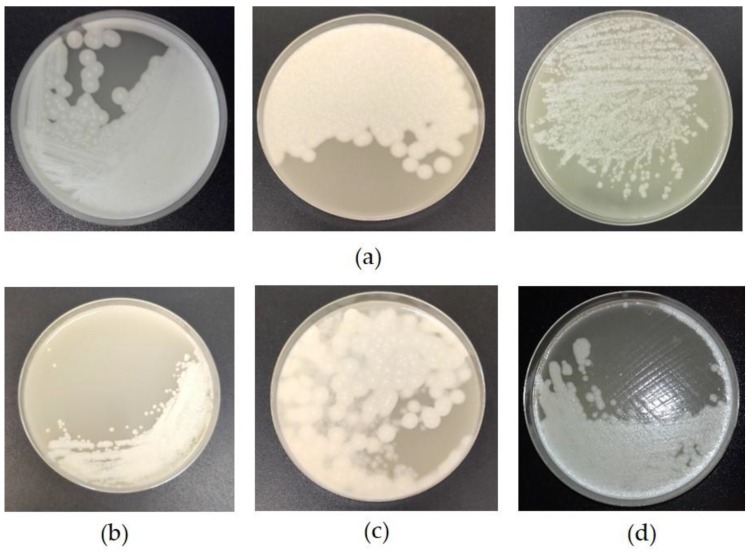
Pictures of colonies of a BoNT type C/D (**a**), D (**b**), D/C (**c**) and nontoxic (**d**) strains plated on Egg Yolk Agar (EYA) after isolation according to the protocol depicted in Figure 3. Note the differences in colony morphology depending on the strain.

**Table 1 toxins-12-00042-t001:** PCR detection of colonies isolated from liver samples spiked with five spores of *C. botulinum* group III per gram of liver.

Spore Reference	BoNT Type	*C. novyi sensu lato* Group Detection after Isolation According to Figure 3	*ntnh* Detection after Isolation According to Figure 3
CIP 109 785	C	+	+
CIP 105 256	D	+	+
B17LNRB4	C/D	+	+
B19LNRB3	D/C	+	+

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
