# Peer review of "Development of An Innovative and Quick Method for the Isolation of Clostridium botulinum Strains Involved in Avian Botulism Outbreaks"

_toxins, 2020, doi:10.3390/toxins12010042_

Round 1
Reviewer 1 Report
In "Development of an innovative and quick method for 2 the isolation of Clostridium botulinum strains 3 involved in avian botulism outbreaks", the authors describe a new technical approach to improve isolation of bacterial isolates expressing BoNT C, C/D, D/C and D. The novelty of the approach relies on the specific ability of C. novyi sensu lato and C. Botulinum spores to survive thermolytic conditions used in the InstaGene method. Overall this manuscript describes a valuable technical advance that improves the isolation and characterization of otherwise difficult-to-culture toxin-producing strains. I do not have any major concerns about the technical data; however, the manuscript could benefit from a restructuring to improve clarity and provide greater context to general readers.
Major concerns: Much of the background and rationale for the study is not addressed until the discussion, leaving readers confused why certain decision were made regarding experimental approaches. The introduction should be revised to provide context for the results section, including (but not limited to) explaining the utility of isolating C. novyi sensu lato as well as C. Botulinum; the role of phage in toxicity; what the current isolation methods are; and how current methods are inefficient or inadequate.
Minor concerns:
1. The purpose of lines 224-227 is unclear and it appears out of place. Should this be part of the introduction?
2. The authors should note that "lowest Ct" represents the samples with the greatest homogeneity
3. Does Figure 1 represent a review of previously published data or newly analyzed data?
Reviewer 2 Report
The authors report a new method for the isolation of C. botulinum group III strains. The article is well written and it provides an improvement of the previous published protocols. Prior publication minor typos should be corrected and the points list below should be addressed:
1) More recent reviews and articles on botulinum neurotoxins mechanism of action and serotypes classification should be citied and discussed. Please refer to: Azarnia Tehran D. et al., Toxins 2018, Pirazzini M et al., Toxicon 2018 and Zhang et al., Nature Comm 2017.
2) The epidemology of botulinum neurotoxins was recently discussed and highlighted in Anniballi F et al., Infect Genet Evol 2018 and Giordani F. et al., Infect Genet Evol 2015. Please update references.
3) Please describe better Figure 1a in the legend. Up to date, it results difficult to understand for the readers. What do colors in the legend mean and represent? Do not report the meaning only in the text.
